# Parliamentary roll-call voting as a complex dynamical system: The case of Chile

Diego Morales-Bader[1,2], Ramón D. Castillo[1]*, Ralf F. A. Cox[3], Carlos Ascencio-Garrido[4]

1 Centro de Investigación en Ciencias Cognitivas, Facultad de Psicología, Universidad de Talca, Talca, Chile, 2 Facultad de Ingeniería y Ciencias, Universidad Adolfo Ibáñez, Santiago, Chile, 3 Department of Developmental Psychology, Faculty of Behavioral and Social Sciences, Heymans Institute for Psychological Research, University of Groningen, Groningen, Netherlands, 4 Escuela de Psicología, Universidad Católica Silva Henríquez, Santiago, Chile

* racastillo@utalca.cl

**Data Availability Statement:** The data underlying the results presented in the study are available from http://opendata.camara.cl https://github.com/dimoralesb/rollcall_votes_chile

## Abstract

A method is proposed to study the temporal variability of legislative roll-call votes in a parliament from the perspective of complex dynamical systems. We studied the Chilean Chamber of Deputies' by analyzing the agreement ratio and the voting outcome of each vote over the last 19 years with a Recurrence Quantification Analysis and an entropy analysis (Sample Entropy). Two significant changes in the temporal variability were found: one in 2014, where the voting outcome became more recurrent and with less entropy, and another in 2018, where the agreement ratio became less recurrent and with higher entropy. These changes may be directly related to major changes in the Chilean electoral system and the composition of the Chamber of Deputies, given that these changes occurred just after the first parliamentary elections with non-compulsory voting (2013 elections) and the first elections with a proportional system in conjunction with an increase in the number of deputies (2017 elections) were held.

## Introduction

It has been proposed that political systems and their behavior are complex systems, and therefore it is important to study them from this perspective [1–4]. Although there is no formal definition of what a complex system is, we may be referring to systems that have several components interacting with each other, that are difficult to predict and model, and that are non-linear, i.e., they do not always respond in the same way to the exact change or with the same intensity, among other characteristics (e.g., emergence). In these systems, sometimes small changes can generate significant changes at a general level, or large changes can sometimes generate no change at all or more minor changes than expected.

Legislative roll-call voting records are frequently used to study a political organization's behavior. These are "yes" or "nay" votes, and depending on the country, there may be abstention votes, as is the case of the Chilean parliament. Legislative roll-call analysis has a long history in political science (i.e., [5]). Usually, this kind of data is used to study the polarization of

**Funding:** RDC and RFAC are funded by the Project for Promotion of International Linkages for Regional Institutions 2022 (FOVI210047), Agencia Nacional de Investigación y Desarrollo de Chile (ANID), and by the Programa de Investigación Asociativa (PIA) en Ciencias Cognitivas (RU-158-2019), Universidad de Talca; DMB is funded by the Doctorate Scholarship FONDECYT-ANID grant # 21220612. The funders had no role in study design, data collection and analysis, decision to publish, or preparation of the manuscript.

**Competing interests:** The authors have declared that no competing interests exist.

political parties [6], their discipline to the party [7], detecting shifts in foreign policy orientations [8, 9], and the influence of controversial votes on elections [10], among other topics. In contrast to these approaches, we propose to study the behavior of legislative roll-call voting variability over time from the perspective of complex systems. Although this approach does not provide us with interpretations as direct as the studies mentioned above, it allows us to visualize changes that may not be detectable with traditional statistical analysis methods.

One could argue that the outcome of a legislative roll-call vote is deterministic and, in many cases, predictable because agreements depend on pre-established alliances, which political party has a majority, and what issue is being voted on, among other factors. We propose that this determinism vanishes if we analyze roll-call votes over a sufficiently long time. By adding temporal dynamics to the analysis, we can find various factors of variability that are difficult to predict, such as emerging changes in the political agenda in response to a crisis, the emergence of controversial issues, changes in political alliances, changes in congressional members due to elections, increase or decrease in the number of parliamentarians, citizens' demands, foreign policy, among several other factors.

From this dynamic perspective, we can understand that systems are constantly changing. This constant change makes it challenging to attribute each variation to a particular situation, mainly due to non-linear changes not directly correlated in time. However, our idea is that, despite local variability, we could analyze the global stability of roll-call votes over time. For this, we will take two variables, the result of each vote (approved or rejected) and the proportion of agreement among parliamentarians in each vote (this variable will be explained in the next section).

Some studies have explored the dynamics of legislative behavior using complex network and social network analysis [11–14]. Given that parliamentarians change in each legislative term, these analyses provide descriptions of local changes. Our proposal seeks to evaluate behavior globally and detect possible significant changes or otherwise to evaluate the stability of the system.

If we treat legislative votes as a time series, where each vote is a time point, we can use an analysis method called Recurrence Quantification Analysis (RQA). RQA is a non-linear method used to discover tenuous correlations and repetitive patterns in a time series [15]. Unlike other analyses, it does not require additional treatment or assumptions about data distribution [16–18]. Recurrence is a fundamental property of complex dynamical systems. It is defined as the ability of a system to return to the proximity of the initial point in phase space. RQA enables us to capture the dynamic organization of such systems based on the amount and patterns of such recurrences in phase space. This technique has been used in various domains to characterize temporal patterns of human and non-human behavior [15, 19–21]. However, despite its use in various disciplines, it has been scarcely used in analyzing political variables. For example, one study found that online social cohesion during the Arab Spring in Syria was related to the frequency of protests [22]. For this, they used a variant of RQA, the Cross Recurrence Quantification Analysis, which analyzes the recurrences of two time series instead of analyzing a time series with itself. In an economic study applying cross-RQA, the non-linear relationship between output and unemployment was studied, as well as the temporal dependence and the existence of state changes. Also, was detected the possibility of predicting transition phases that coincided with periods in which the economy suffered recessions [23]. Both studies show that the RQA could be useful for detecting patterns and changes in a time series that may be difficult to analyze with traditional statistical methods.

An additional useful metric of systems dynamics analysis is entropy. Entropy can be defined as the amount of information in a time series or signal. The more deterministic a signal is, it will contain less information. Sample Entropy (SE) is a typical analysis to study the

randomness or stability of a signal [24]. SE can discriminate stochastic processes and noisy deterministic datasets [25] and is nearly unaffected by low-level noise [26]. SE estimates the randomness of a time series without any previous knowledge about the source generating the data [27]. It measures how much a given data point depends on the values of past data points. Lower values indicate more self-similarity of the time series (approximately similar to a part of itself), while higher values indicate a less predictable series. An example of entropy in political analysis is the study by Marmani et al. [28], who used entropy and the Gini index to analyze municipal and parliamentary voting in Italy. They found high entropy levels but with variations in some election periods.

In this study, we considered the roll call votes of the Chilean Chamber of Deputies. We argue that non-linear time series analysis techniques, such as those described above, could provide new insights into the voting dynamics and the system's behavior [29, 30].

The Chilean political system is a multiparty system where the president is the head of state and head of government. Legislative power is exercised by two chambers of the national congress (Chamber of Deputies and Senate) and by the republic's president. The functions of the Chamber of Deputies are creating and approving laws, which it performs jointly with the Senate. It can supervise the acts of the government. It is currently composed of 155 members, directly elected from 28 electoral districts nationwide for four years.

Deputies can express their vote in three ways; voting for, against, or abstaining. There are no obstruction mechanisms, and all absences must be justified. However, a mechanism known as "pairing" allows deputies not to participate in votes. Pairing is an agreement between two deputies from different benches or committees, agreeing not to participate in any vote if one is absent for a certain period.

Each legislative term lasts four years, and all members of the Chamber of Deputies are renewed through elections usually held in conjunction with the presidential elections. Deputies can be reelected for a maximum of two terms.

The board of directors of the Chamber of Deputies is composed of a President, a First Vice President, and a Second Vice President. An absolute majority of the deputies elect the board of directors by secret ballot. The President and Vice-Presidents of the Chamber can be reelected. The board of directors serves until the end of the legislative period; however, there are usually alternation agreements between the political parties to elect and occupy the chamber's presidency for short periods. For example, there may have been four chamber presidents, each occupying the seat for one year in one legislative term.

All members of the chamber's board, including the president, can vote in the sessions. The functions of the president include presiding over the sessions and directing the debates, taking care of the chamber's rules of procedure, opening, suspending, and adjourning sessions, declaring the inadmissibility of projects and indications following the Organic Constitutional Law of the National Congress, among other functions.

In the context of the Chilean presidential regime, where the figure of the president has broad powers and little external control, the effective power possessed by both chambers of congress in terms of determining the agenda and the political life of the country is lower compared to the rest of Latin America [31]. Nevertheless, the role of the chamber is fundamental to understanding the development and projections of Chilean politics, where legislative productivity has been progressively increasing [32]. For example, productivity increases consistently in the first half of the presidential term, given the thrust of the incoming government to generate reforms and laws associated with its government plan, which gradually decreases over the last two years of administration [33].

Since 1990, the election system of parliamentarians was a binomial system with compulsory voting based on lists of candidates, which tended to overrepresent the second majority. This

electoral system implied a balanced composition between two major political forces, leaving out the rest of the political movements in a system that could be called forced bipartisanship, making the political parties have as main competition their own allies in the list presented for the elections [34].

A change to the electoral system was approved in 2012 when voting became non-compulsory (the first election with this change was in 2013). Then in 2015, the binomial system was replaced by a proportional election system following the D'Hont model. In addition, the districts and their sizes were reorganized, increasing the number of representatives from 120 to 155. This change was implemented for the first time in the elections of 2017. This new electoral system generates greater representativeness in the overall composition of the chamber, but with the cost of greater difficulty for the executive power to achieve stable governance over time [35].

All the characteristics of the Chilean political system described above could set the conditions for complex behavior. In this context, the goal of this study is to explore whether some non-linear analysis methods can detect significant changes over time in the legislative work of the Chamber of Deputies of Chile with some complexity measures (recurrence and determinism from the RQA, and entropy from Sample Entropy). We will use the roll-call votes recorded from 2002 to the end of 2021. More specifically, generating two variables from the roll-call votes data; the agreement ratio (number of votes of the majority option divided by the total number of votes) and the voting outcome of each vote (approved or rejected).

## Methods

### Dataset

A total of 19566 legislative votes of the Chilean Chamber of Deputies from 2002-03-19 to 2021-12-22 were obtained (openly available at http://opendata.camara.cl). The dataset initially had 19597 votes, but 31 were omitted due to record errors. By reading the session logs, other votes with recording errors were manually corrected when possible (n = 36). The data was downloaded and processed with a custom Jupyter Notebook, which can be downloaded at the following link: https://github.com/dimoralesb/rollcall_votes_chile.

The data contains the number of deputies who voted in favor, against, or abstained, the result of the vote, the type of vote (bill, agreement, resolution, or others), the quorum required (simple quorum, 2/3, 3/5, among others), and the date of the vote.

From the data, we calculated the agreement ratio of each vote as a measure of the variability of the agreements. The agreement ratio is obtained by dividing the number of votes of the majority option (in favor or against) by the total number of votes.

The minimum agreement ratio was 0.30 (more abstentions than "Yes" or "No" votes), and the maximum was 1 (mean = 0.83, SD = 0.19).

The second variable, the voting outcome, is obtained directly from the data. Initially, the data includes some voting results labeled as a tie (n = 22). Since ties imply that the outcome was a rejection, these records were re-labeled as rejected. Consequently, votes labeled as "no quorum" (n = 371) were also classified as rejected according to the rules of the Chamber of Deputies. Then, a numerical recode was made, where value "1" was assigned to approved votes and value "2" to rejected votes.

### Data analysis

First, descriptive analyses were performed on the agreement ratio, voting outcome, and total votes per year.

For the non-linear data analyses, the data were divided into each legislative period, performing an RQA and SE for each. Then, for a second analysis, the entire data series was divided into (partly overlapping) windows of 256 data points with steps of 10 points (i.e., window 1: votes 1 to 257, window 2: votes 11 to 267, and so on). In each window, an RQA and a SE were performed.

These fixed-size windows include different time ranges since a variable amount of time may elapse in those 256 votes. For example, one window may occur in three months, while another may represent four or five months of data. The purpose of dividing the data into smaller, partially overlapping windows is to observe variations in the variables studied in greater detail.

To compare the results without the effect of the time structure, the order of the original series was randomized 30 times [15, 19]. Then, the non-linear analyses were performed for each randomization (RQA and SE). A mean and a confidence interval were calculated for each window and legislative period.

We used an algorithm to detect change points in the windowed analyses of our measures to determine when the system significantly changed (change in mean). We aimed to detect the most critical change point in the system since, due to the nature of the data, it could be considered that there are many change points in the system. The R-package "cpm" was used [48]. The Mann-Whitney test statistic with a p value $< = 0.01$ was selected for change point detection.

The recurrence quantification analysis and the Sample Entropy are described below, along with their important parameters.

### Recurrence Quantification Analysis (RQA).

RQA [20, 36–38] was performed for the agreement ratio and voting outcome. Since recurrence is a property of dynamic systems, RQA allows quantifying the number and duration of the recurrences in phase space. RQA is based on procedures for visualizing recurrence patterns in dynamic systems named recurrence plots (RP). In general terms, RP is a two-dimensional representation of a multidimensional phase space trajectory $\vec{x}$. RPs show the geometric and dynamic properties of the temporal trajectories that a system can generate in its original multidimensional space [39]. RPs visualize the recurrence of a state at time $i$ at a different time $j$ in a two-dimensional squared matrix with ones and zeros (or black and white dots in a plot), where both axes are time axes (S1 Appendix). RP can be mathematically expressed as:

$$R_{i,j} = \Theta\left(\varepsilon - \parallel \vec{x_i} - \vec{x_j} \parallel\right), \; x_i \in \mathbb{R}^m, \; i,j = 1,\ldots,N, \tag{1}$$

Where $N$ is the number of considered states of $x_i.\varepsilon$ is a threshold distance, $\parallel\cdot\parallel$ is a norm (usually the Euclidean norm), and $\Theta(\cdot)$ is the Heaviside function, which is 1 when $x_i \approx x_j$, and 0 otherwise. $x_i \approx x_j$ means equality up to an error or distance threshold $\varepsilon$. In short, the matrix tells us when similar states of the underlying system occur.

RQA can reconstruct the phase space of a multidimensional system based on Taken's embedding theorem [40]. This theorem proposes that under some conditions, it is possible to roughly reconstruct the state-space of a dynamical system by delay-embedding only one of its time series as follows:

$$\vec{x_i} = \left(u_i, u_{i+d}, \ldots, u_{i+d(m-1)}\right) \tag{2}$$

Where $u_i$ is the time series, $m$ is the embedding dimension, and $d$ is the time delay. The delay parameter helps to retrieve the m-dimensions using the delayed embedding method. This parameter considers the properties of the sample of observations, finding points in the time

series that can be used to reconstruct the latent dimensions. The embedded dimensions ($m$) correspond to an estimate of the latent dimensions that would configure the observed system. Finally, the radius ($\varepsilon$) specifies an interval in which two values are considered equal. In general, the radius is not easy to determine in a continuous series, but it is suggested to set a value that allows counting between 1% to 5% of recurrences [41]. The number of recurrences is related to the established radius, so if it is high (greater freedom), the percentage of recurrence points will be greater.

We used the Average Mutual Information (AMI) to estimate the delay to use. AMI helps to quantify the amount of knowledge gained about the value of $x(t+d)$ when observing $x(t)$. AMI creates a histogram of the data using bins. Let $p_i$ be the probability that the signal has a value inside the $i^{th}$ bin and let $p_{ij}(d)$ be the probability that $x(t)$ is in bin $i$ and $x(t+d)$ is in bin $j$. Then, AMI for delay $d$ is defined as:

$$AMI(d) = \sum_{i,j} p_{ij} \log\left(\frac{p_{ij}}{p_i * p_j}\right) \tag{3}$$

Next, to estimate an adequate number of embedded dimensions, we used the False Nearest Neighbor (FNN) algorithm [42, 43]. The goal of this method is to determine the minimal sufficient embedding dimensions. If a series is not adequately embedded with its true dimensionality, there is a risk of classifying false recurrences. This method assumes that two points in a time series that are near to each other in the sufficient embedding dimension ($m$) should remain close as the dimension increases. If the reader is interested in how to calculate this iterative method, we suggest reviewing the references mentioned here [42, 43].

Conventionally, the first minimum of the Average Mutual Information (AMI) and the False Nearest Neighbor (FNN), or the point at which those functions level-off, are indicative of the optimal delay and embedding dimension [16, 44]. We used the R-package "tseriesChaos" [45], which contains an implementation of the AMI and the FNN methods to estimate the delay and the embedding dimensions for the RQA.

To perform the RQA we used the python package "PyRQA" [46]. First, we estimated the agreement ratio's RQA parameters (delay, embedding dimensions, and threshold). In order to identify the optimal delay for the agreement ratio, the median of the first minimum value of the AMI of each period and window was used. For the embedding dimensions, the median of the global minimum of the FNN was used. These functions indicated that a delay of 1 and an embedded dimension of 6 was optimal for analyzing legislative periods and time-varying windows.

For selecting the threshold, a general guideline is that the percentage of recurrence points (REC) should remain low but not so small as to produce a floor effect with values of REC near or at 0.0 [17, 41]. The threshold value of $\varepsilon = 0.17$ was chosen to yield between 1% and 5% (4% average) recurrence points for the agreement ratio.

For categorical variables like the voting outcome, a delay and embedding dimension of 1 and a small threshold of 0.0001 is often sufficient [19, 30]. Therefore, estimating these parameters with the methods indicated above is not necessary.

Several measures can be obtained from the RQA. For this study, we will consider two; the recurrence ratio (RR) and the determinism (DET). RR indicates the probability that a specific state will recur, while DET relates to the predictability of the system [16]. Mathematically, RR is the density of recurrence points in a recurrence plot, while DET is the percentage of recurrence points that form diagonal lines in the recurrence plot of minimal length $\ell_{min}$.

Considering the Eq 2, REC and DET are estimated as follows:

$$RR = \frac{1}{N^2}\sum_{i,j=1}^{N} R(i,j) \quad DET = \frac{\sum_{\ell=\ell_{min}}^{N} \ell P(\ell)}{\sum_{\ell=1}^{N} \ell P(\ell)} \tag{4}$$

For DET, $\ell_{min}$ was set to 6. This number corresponds to the median number of votes parliamentarians perform per day.

**Sample entropy (SE).** SE is a complexity measure that quantifies the system fluctuations' degree of regularity and unpredictability [24]. Mathematically is defined as the negative natural logarithm of the conditional probability that two sequences similar of $m$ points remain similar at the next point (excluding self-matches). Therefore, a lower value for the SE corresponds to a higher probability indicating more self-similarity. SE is computed as follows: from a vector $X_N$, two sequences of $m$ consecutive points $X_m(i)$ and $X_m(j)$ are selected to compute the maximum distance and compared to tolerance $\gamma$ for repeated sequences counting. For the sequence $X_m(i)$ its count is defined as $B_i^m(\gamma)$. The maximum distance is computed as follow:

$$d(X_m(i), X_m(j)) = \max\Big(|x_i + k, x_j + k|\Big) \le \gamma(k \in [0, m-1], \gamma \ge 0) \tag{5}$$

Where the tolerance $\gamma$ equals to $0.2 * SD(X_N)$ in our case. $B^m(\gamma)$, is the average amount of counts $(B_i^m(\gamma))$, · and · $B^{m+1}(\gamma)$ is the average of $m+1$ consecutive points. Then, SE is computed as follows:

$$SE(N, m, \gamma) = -\ln\left(\frac{B^{m+1}(\gamma)}{B^m(\gamma)}\right) \tag{6}$$

The Python package "nolds" [47], which implements this method, was used with the default parameters to compute the SE.

## Results

### Descriptive analysis

Descriptive analyses indicated that votes usually have a high agreement ratio among parliamentarians (Fig 1, panel B) and a high approval rate (Fig 1, panel C). Besides, the votes per year (Fig 1, panel A) had increased in an approximately linear way ($R^2 = 0.80$) from 2002 (n = 454) to 2021 (n = 1267), dropping almost consistently on election years except for 2013 (election years: 2005, 2009, 2013, 2017 and 2021).

Next, the data was divided into legislative periods to observe their variation over time. We found that the percentage of approved votes was higher in the last two legislative periods (Table 1). The 2010–2014 term had the lowest percentage of approved votes (76.0%), whereas the 2014–2018 term had the highest (88.6%). Also, we found that the agreement ratio of approved and rejected votes was lower in the last two legislative periods (Table 1). In other words, more votes were approved in the last two terms than in the previous terms, but with a lower agreement ratio among parliamentarians.

### Nonlinear analysis

We split the time series (agreement ratio and voting outcome) into legislative periods to study potential changes in the voting behavior and political system between these periods. We performed the RQA (taking the recurrence rate and determinism) and SE on these periods for both variables. Subsequently, we randomized the order of all points in the time series in each period 30 times (i.e., random shuffling) [15, 19]. We performed the same analyses to compare

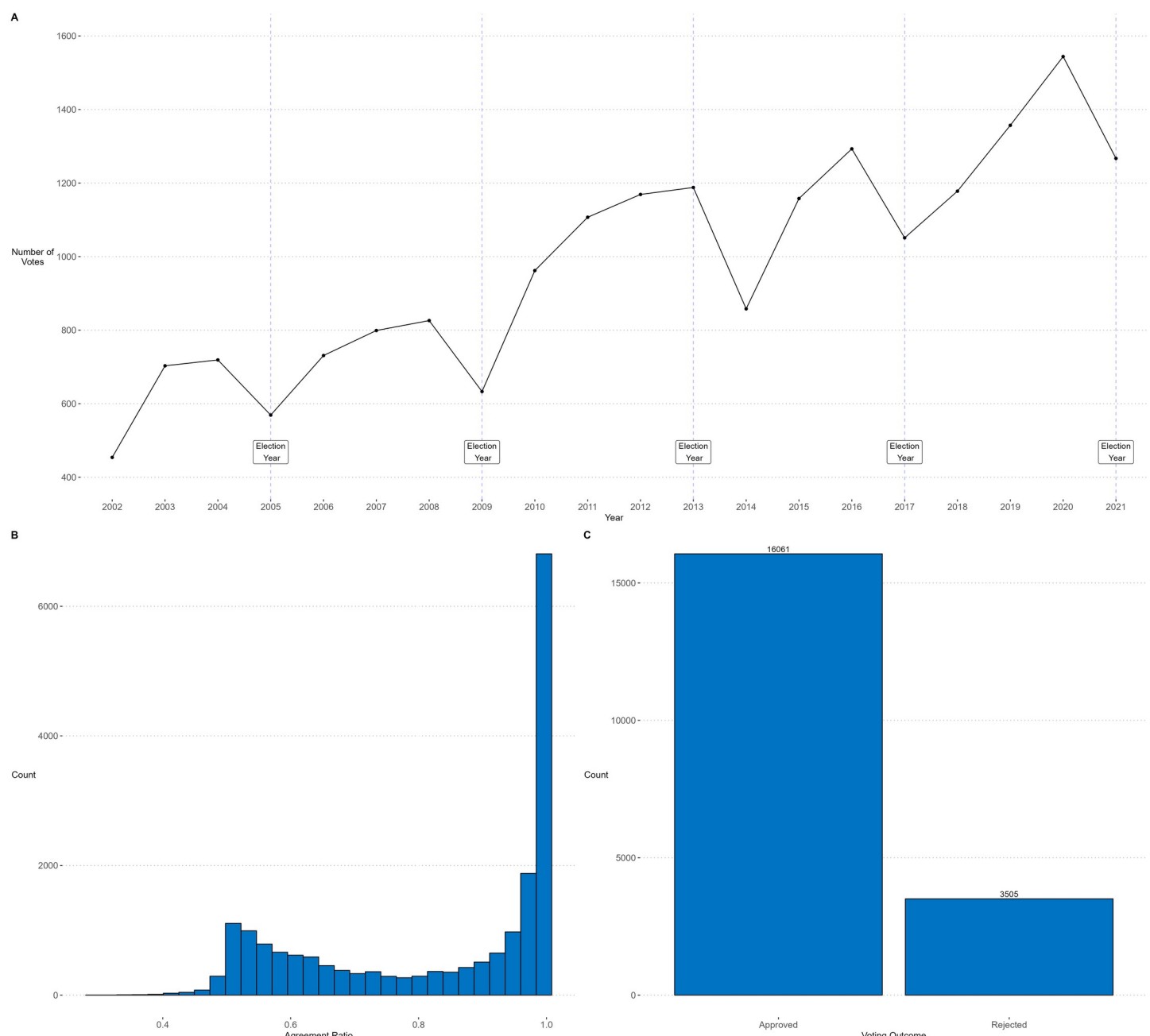

**Fig 1.** Number of votes per year from 2002 to 2021 (Panel A), Histogram of agreement ratio from 2002 to 2021 (Panel B), and frequency of voting outcome from 2002 to 2021 (Panel C).

the resulting measures with those of the original time series. If the data were stochastic, these analyses should indicate a similar result with the original data series and randomly shuffled data. Given the nature of our data, there should be differences from the randomized series.

The results of the RQA indicate that the recurrence and determinism of the agreement ratio were relatively stable in the first four legislative periods (Fig 2, Panel A and B). However, since 2018, a lower recurrence ratio and determinism were found, much closer to the

**Table 1. Percentage of approved and rejected votes and average agreement ratio in each legislative period from 2002 to 2021.**

| Legislative Period | Approved | Mean Agreement Ratio of Approved Bills | Rejected | Mean Agreement Ratio of Rejected Bills |
|---|---|---|---|---|
| 2002–2006 (N = 2505) | 80.60% | 0.89 (SD = 0.16) | 19.40% | 0.69 (SD = 0.18) |
| 2006–2010 (N = 2990) | 80.90% | 0.89 (SD = 0.16) | 19.10% | 0.66 (SD = 0.17) |
| 2010–2014 (N = 4461) | 76.00% | 0.87 (SD = 0.17) | 24.00% | 0.73 (SD = 0.21) |
| 2014–2018 (N = 4464) | 88.60% | 0.87 (SD = 0.15) | 11.40% | 0.63 (SD = 0.16) |
| 2018–2021 (N = 4516) | 83.10% | 0.82 (SD = 0.18) | 16.90% | 0.57 (SD = 0.11) |

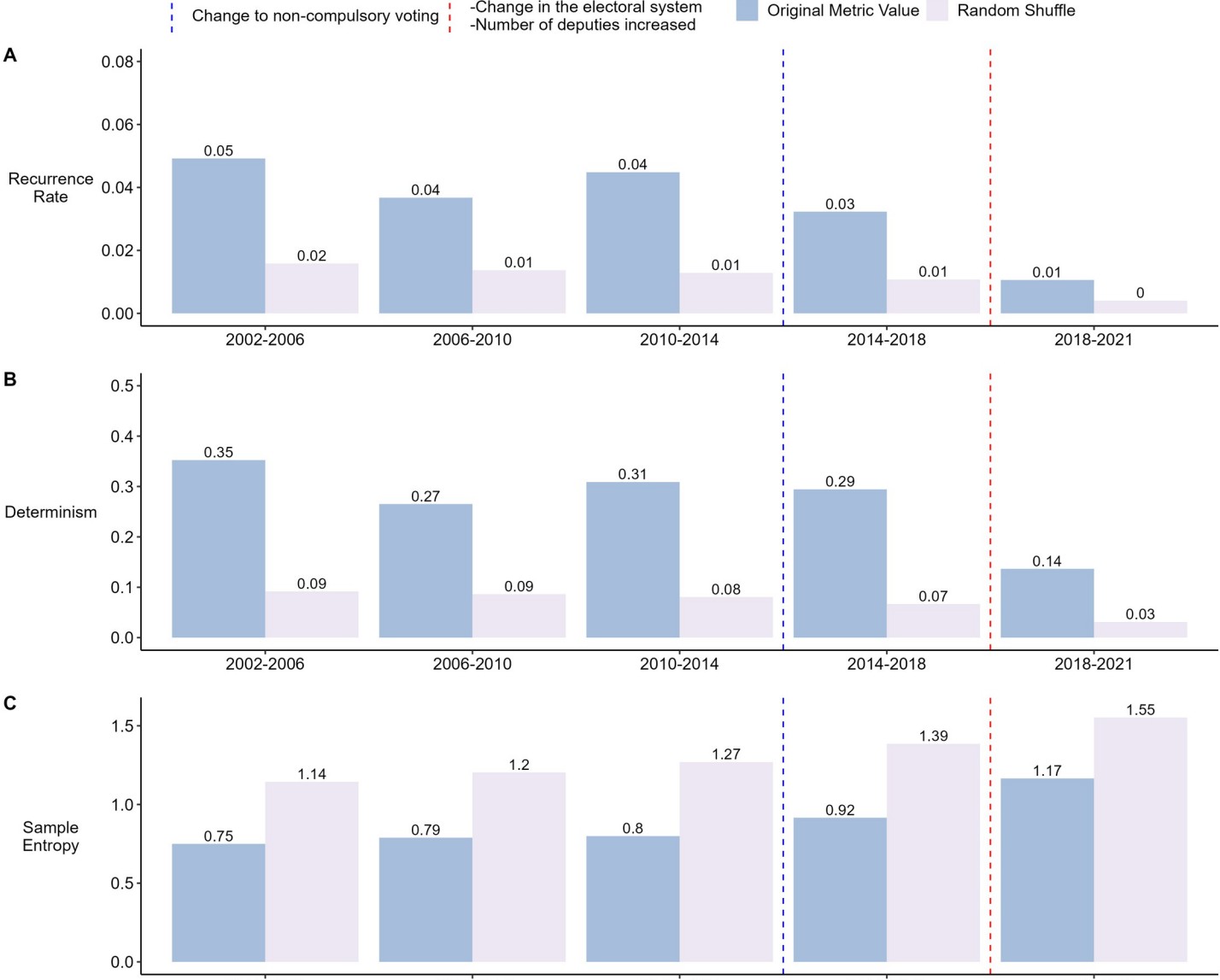

**Fig 2. Analysis of the Agreement Ratio for each legislative period.** RQA is used to obtain the Recurrence Rate (Panel A) and Determinism (Panel B). Sample Entropy is used to measure the entropy of the data (Panel C). For each metric, the result obtained with the original data series is shown, along with the average result of randomly shuffling the data for each period 30 times and performing the analyses on each shuffle. The blue dashed vertical line shows when the first parliamentary elections with non-compulsory voting took place. The red dashed vertical line shows the first parliamentary elections with a propositional election system and the increase in the number of deputies.

randomization than in previous years. This variation would indicate a change in the system, being less predictable and less likely for the same state to be repeated than in previous periods.

The SE shows that entropy has increased in each legislative period, with a more noticeable increase in the last period (Fig 2, Panel C). In other words, the unpredictability of the agreement ratio has been increasing, especially in the last legislative period. This coincides with the findings of the RQA, where the last period marks a different trend with respect to the previous periods. It is important to note that the last legislative period coincides with the change in the parliamentary electoral system and the increase in the number of deputies.

On the other hand, the analysis of the voting outcomes (Fig 3, Panel A and B) showed a relatively stable recurrence and determinism in the first three periods, with a decrease from 2010–2014, before a notable increase in the 2014–2018 term. While the last term showed a lower recurrence and determinism than the previous one, it is still higher than the first three periods. These results would indicate a change to the system contrary to that observed in the agreement ratio. The system becomes less recurrent and predictable from one term (2010–2014), then becomes a more deterministic system susceptible to visiting the same states in the last two periods.

It is observed that when each period is randomized, the recurrence ratio remains the same. This is because recurrence quantifies the number of events. For categorical variables, the number of events remains the same when randomizing the order of the data within the same period or window. This is not the case for the windowed analysis because we shuffled the order of the whole series.

The SE follows the trend found by the RQA for the voting outcome. The unpredictability increased in 2010–2014 and decreased significantly in 2014–2018. Since 2018 the sample entropy has grown again but remained lower than the first three terms. These results indicate a more deterministic system in the last two terms (Fig 3, Panel C). This change in the system occurred during the period when for the first time, deputies were elected in non-compulsory voting elections.

Next, to explore the changes and transitions in the system in more detail, we performed the RQA and SE by partially overlapping windows across the entire data. Again, the time series were shuffled 30 times, and the same analyses were performed. A change-point algorithm [48] was used to detect the most significant changes in the time series using the Mann-Whitney test statistic with a p-value $< = 0.01$.

The RQA of the agreement ratio (Fig 4, Panel A and B) shows a decay in recurrence rate and determinism in the last legislative periods. For the recurrence ratio, a change point occurs at window 1453 ($p < 0.01$), and for determinism, the change occurs at window 1456 ($p < 0.01$), both shortly after the beginning of the 2018–2021 term. Visual inspection suggests that both indicators are closer to random levels in the last period and sometimes below, more frequently than in previous periods.

The SE is consistent with the findings of the RQA. The algorithm indicated a change after the beginning of the 2018–2021 term at window 1462 ($p < 0.01$). It is also observed that the system's unpredictability increases in the last period, surpassing the randomized series line for a moment (Fig 4, Panel C).

The change points detected in each metric are close to each other. They also occur after the change in the electoral system from a binomial to a proportional one and after the increase in the number of deputies.

For the voting outcome, the RQA showed a change point ($p < 0.01$) near the beginning of the 2014–2018 term (Fig 5, Panel A and B). After this point, the data series increases in recurrence ratio and determinism. Visually we observe the decay of both indicators before the change point occurs, signaling the increase in the values of the series. While both series

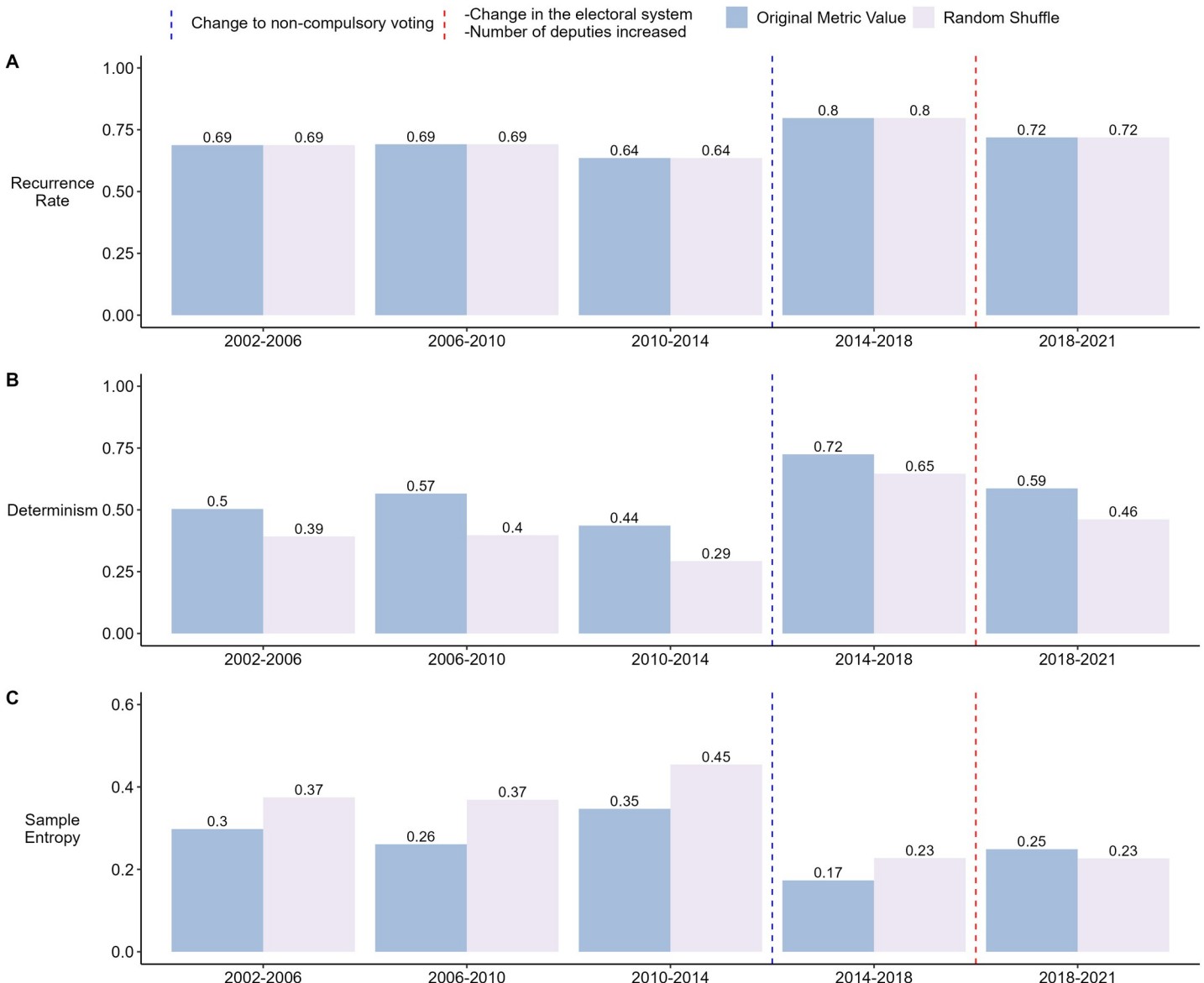

**Fig 3. Analysis of the voting outcome for each legislative period.** RQA is used to obtain the Recurrence Rate (Panel A) and Determinism (Panel B). Sample Entropy is used to measure the entropy of the data (Panel C). For each metric, the result obtained with the original data series is shown, along with the average result of randomly shuffling the data for each period 30 times and performing the analyses on each shuffle. The blue dashed vertical line shows when the first parliamentary elections with non-compulsory voting took place. The red dashed vertical line shows the first parliamentary elections with a propositional election system and the increase in the number of deputies.

oscillate between the random lines in the first periods, the series remains below this line before the change point. After the change point, the series remained above the random line more frequently than in other legislative periods.

Again, the SE is consistent with the RQA. The unpredictability increased slightly in 2010–2014 and decreased significantly in 2014–2018 (Fig 3, Panel C). Since 2018 the sample entropy has grown again but remained lower than the first three terms. These results indicate a more deterministic system from 2014 to 2021. The windowed analysis shows this trend in greater detail (Fig 5, Panel C). First, the series fluctuates above and below the randomized line. For a moment, within the 2010–2014 term, it stays above the randomized line and decreases notably

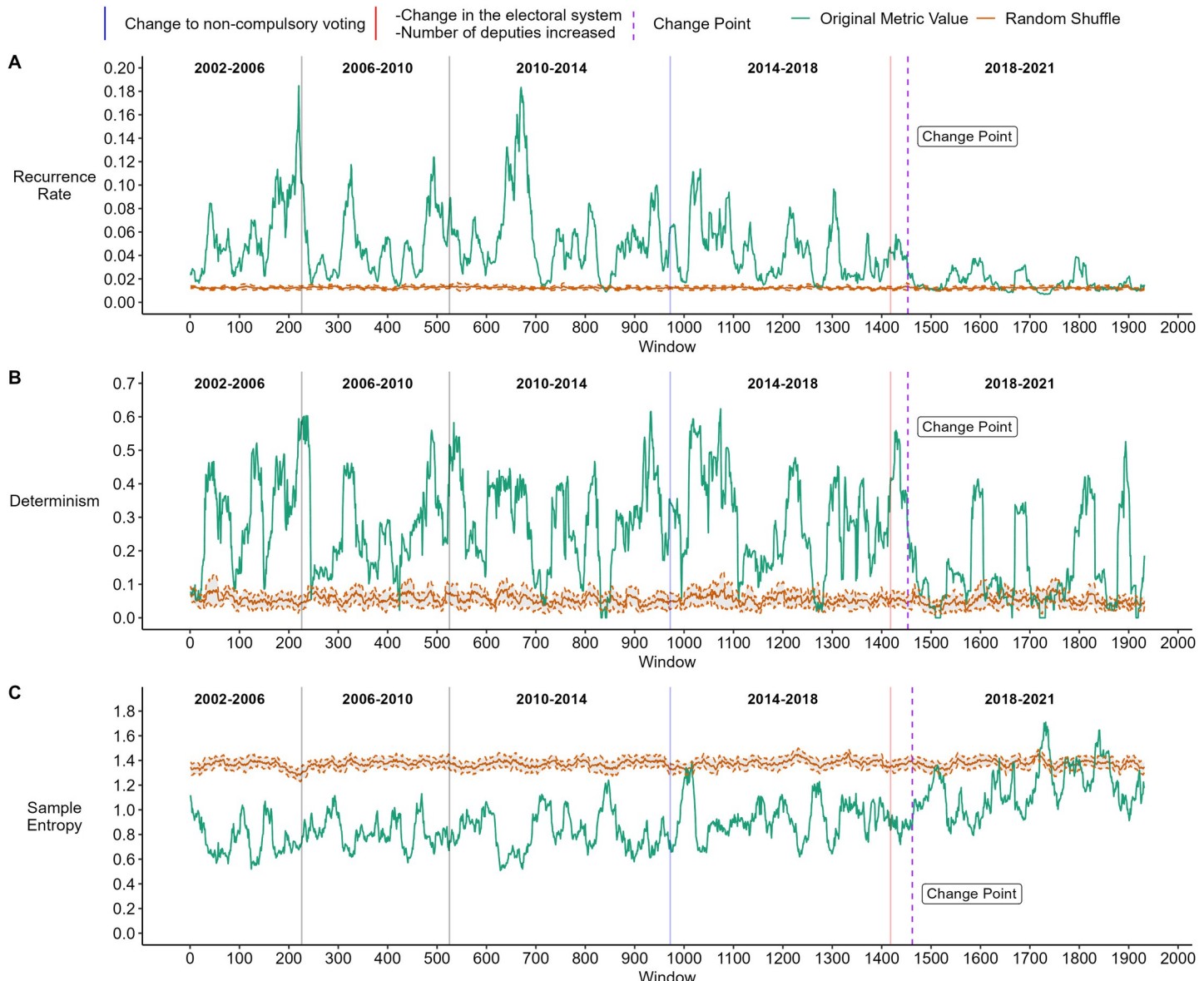

**Fig 4. Analysis of the agreement ratio by partially overlapping windows of 256 data points with steps of 10 points.** RQA is used to obtain the Recurrence Rate (Panel A) and Determinism (Panel B) for each window. Sample Entropy is used to measure the entropy of the data in each window (Panel C). For each metric, the result obtained with the original data series is shown in green, along with the average result of randomly shuffling the data 30 times and performing the analyses on each shuffle (red horizontal line). In total, 1932 windows were analyzed. The blue vertical line shows when the first parliamentary elections with non-compulsory voting took place. The red vertical line shows the first parliamentary elections with a propositional election system and the increase in the number of deputies. The purple dashed vertical line indicates the change point detected with the R package algorithm "cpm" with p < = 0.01.

from window 990 (p < 0.01), corresponding to the beginning of the 2014–2018 term. Finally, SE increases slightly towards 2018–2021 but without resembling the levels of the first periods.

As was the case with the results of the agreement ratio, in the result of the voting outcome, the change points also occurred around the same point. However, unlike what was found in the agreement ratio, here, the change point occurs after the first elections with non-compulsory voting were held. However, we can see that before this, there seems to be a distinct variability for a moment in all three metrics. This change point will be discussed in more detail in the next section.

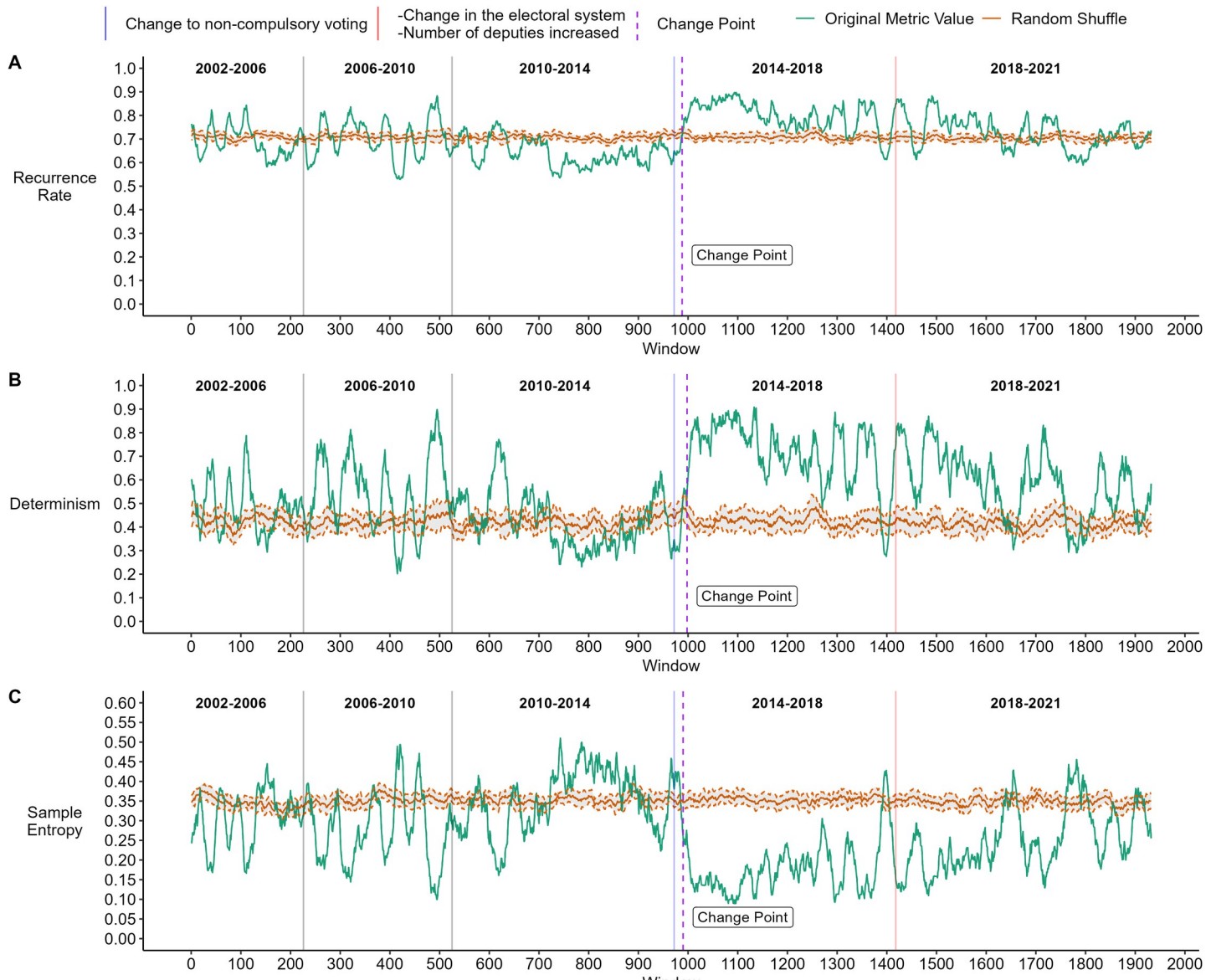

**Fig 5. Analysis of the Voting Outcome by partially overlapping windows of 256 data points with steps of 10 points.** RQA is used to obtain the Recurrence Rate (Panel A) and Determinism (Panel B) for each window. Sample Entropy is used to measure the entropy of the data in each window (Panel C). For each metric, the result obtained with the original data series is shown in green, along with the average result of randomly shuffling the data 30 times and performing the analyses on each shuffle (red horizontal line). In total, 1932 windows were analyzed. The blue vertical line shows when the first parliamentary elections with non-compulsory voting took place. The red vertical line shows the first parliamentary elections with a propositional election system and the increase in the number of deputies. The purple dashed vertical line indicates the change point detected with the R package algorithm "cpm" with p < = 0.01.

In both Figs 4 and 5, we notice that the metrics constantly vary. This variability may be natural and expected, given that the votes are constantly moving due to different factors. These analyses aim to detect when this variability significatively changes and when the system transitions to a new state. For this reason, the analyses are not focused on small local changes with small time scales (a few months in this context) but on changes that move the system's structure on larger time scales (years or legislative periods).

Overall, we found a significant change in the agreement ratio at the beginning of the last legislative period. In this period, the temporal variability of the agreement ratio is less

recurrent and less stable than in previous periods. In addition, the average agreement ratio is the lowest in this period. This change could be related to an important change in the chamber of deputies since it was in that period when the number of parliamentarians increased, and the election system changed from a binomial to a proportional one. We did not find a significant change in the voting outcome in that period. However, we detected a significant change at the beginning of the 2014–2018 term. From that point, the variability of voting outcomes tends to be more predictable and stable compared to previous periods. In this period, the percentage of ballots approved reached its highest point (88.6%). This change seems to match with the first parliamentary elections with non-compulsory voting. However, in the next section, another hypothesis is explored.

## Discussion

In this study, we analyzed 19 years of the temporal variability of the agreement ratio and the voting outcome of the Chilean Chamber of Deputies using non-linear analysis techniques. We argued that these techniques could be useful for detecting significant global changes in the temporal variability of the data. While voting is always in constant change due to various factors, both internal (normal functioning of the system) and external (crises, protests, emerging issues, policy changes, among others), these techniques could help us to detect changes that move away from the traditional states. Indeed, we detected two significant changes, one related to the voting outcome and the other related to the agreement ratio.

The first change detected indicates that from 2014 onwards, voting results become more stable and deterministic compared to previous periods. This change matches the first parliamentary votes with non-compulsory voting. In this period, the highest percentage of approved votes is reached.

One of the reasons that justified the reform of automatic registration and voluntary voting was the aging of the electoral population, which originated mainly from the low registration of young people in the electoral registers. The reform reduced the age bias; however, in the first elections under this modality, the percentage of voter turnout increased by 38% compared to the 2009 elections.

We found that in this period (2014–2018), the percentage of deputies that were government supporters (S2 Appendix) was higher than in other periods (59.2%), which would explain why the percentage of approved votes was higher, with the system becoming more recurrent and stable. However, even though recurrence and determinism decreased, and entropy increased in the next period, they are still above the previous periods even though the percentage of deputies supporting the government (S2 Appendix) returned to usual percentages (close to 50%). This would indicate that the change detected is not focused only on the 2014–2018 term. An evaluation of the 2022–2026 term could confirm whether this change was global or local.

The next step would be to explore what was happening politically in that period. However, we have another hypothesis to explain this change from the complex systems framework. This change could represent a phase transition related to another variable. Phase transitions are a common feature of complex dynamical systems representing a (sudden) change in the global state of the system when a threshold or critical parameter value is reached [41, 49, 50]. We believe this critical parameter could be the number of votes cast by the deputies. In Fig 1, we saw that the number of votes per year increased almost linearly, only decreasing in election years. This observation coincides in part with Badillo et al. [33], who found that the parliament's productivity decreases gradually during the last two years of its term; however, they do not mention such a marked decrease in the election years we found in this study.

Coincidentally, before the change we detected occurred, the number of votes did not decrease in the election year but decreased the following year.

It is clear that the number of votes per year cannot increase unlimitedly, nor does the time of the debate and voting periods increase to meet the demand; therefore, the system must adapt to these changes. This change may be a result of that adaptation, where the voting outcome had to become more stable to respond to the demands of the environment. This can be represented in legislative sessions with the approval or rejection of various items simultaneously on which there is relative consensus without going into greater detail. It is important to mention that the bills were previously worked on by commissions formed by a group of deputies. They are in charge of preparing and discussing these documents with their political parties, so not all of them require further debate.

Furthermore, this hypothesis could explain why, before this significant change occurred, an increase in entropy and a decrease in recurrence and determinism were detected in the window analysis. This could indicate an early warning before the system transitions to a new phase [51].

While these methods, as we apply them here, do not allow us to establish causal relationships, they help us to know where to focus our attention, so this significant change detected may be the starting point for future research.

The second significant change detected related to the agreement ratio occurred at the beginning of the last period analyzed (2018–2021). In this period, the temporal variability of the agreement ratio became less recurrent and less deterministic, with more entropy, i.e., it became less regular. This change occurred after the electoral system changed from a binomial to a proportional one [52].

This electoral change introduced new political forces into parliament, which might have been more difficult under the binomial system. By introducing new political forces, the dynamics of parliament are likely to change, making it more difficult to reach agreements as there are now more political blocks to negotiate. We cannot attribute a causal relationship to the change found in 2018 because several factors affect the parliament's behavior. Nevertheless, introducing new political forces is a significant organizational change in the system, bound to affect its dynamics. In addition, the 2017 elections also saw an increase in deputies from 120 to 155. We believe this increase also benefited the entry of new political forces into congress.

There is a difficult coexistence between presidential regimes and multi-party systems, which present fewer incentives for party discipline and loyalty, forcing the executive to negotiate permanently to achieve governability [53]. This is in line with what is observed in our results since it introduces an additional source of variability. However, this instability would be typical of the combination of presidential regimes such as Chile's and disaggregated party systems, which should stabilize as a more moderate presidential system is established [35]. Our results could be evidence of how a change in the election system can produce significant changes in the functioning of a parliament. However, more evidence from other parliaments is needed to support this idea.

It is important to note that in the 2018–2021 term, several changes and fluctuations in the national political environment occurred. In October 2019, a social outbreak and a political crisis, and in 2020 the coronavirus pandemic emerged. These events affected the priorities in the government plans and altered the legislative agenda of the Chamber of Deputies. Despite these changes, the system still exhibits recurrence patterns that distinguish it from a fully entropic system. A complementary analysis using a technique called Detrended Fluctuation Analysis (DFA) found that despite all these changes detected at large and small scales, the system does not lose its self-similarity [54, 55], i.e., the votes continue to maintain a certain structure at different time scales (S3 Appendix).

In this study, we rely on the idea that the temporal variability of voting has features of complex systems, even though without this time scale, voting may be deterministic. To test this idea, we performed a series of complementary analyses following the method of Olthof et al. [29], which evaluates different characteristics of complex systems (Dependency on past values, long-range and non-stationary temporal correlations, regime shifts, and sensitive dependence on initial conditions). We found that our two variables meet the expected properties of complex systems (S4 Appendix). Moreover, in Fig 1, we see a high probability that the agreement ratio is over 90% (panel b) and a high probability that a vote will pass (panel c). Despite this, by including time as a variable, we found distinct temporal patterns even with data that, at first glance, have little variability.

According to the findings of this study, we believe that these methods are useful to describe a political organization such as the parliament through the historical record of roll-call votes and, at the same time to detect relevant changes over time. This method could be replicated on voting data from any national or supranational parliament with a historical voting record. A comparative study of the dynamic characteristics of parliaments with different election systems and different numbers of political parties involved (two-party vs. multi-party systems) could lead to valuable insights into the organization and behavior of political systems. Additionally, studying the time gap between each vote in the legislative periods, similar to how interstimulus intervals (ISI) have been studied [56], could provide more details on how parliamentary sessions behave and evolve. For example, it would provide information on how chaotic or structured legislative sessions are. Also, specific questions about the relationship between chaotic moments in roll-call voting behavior and particular events in national politics could be addressed.

A limitation of this study is that, given its descriptive approach, it is impossible to establish direct relationships between national policy events and the changes found in this study. We can only make conjectures, given the association of the timing of these changes with the timing of the events. Several events and variables may co-occur with internal and external influencing factors. These factors influence in different ways depending on the strength of the external influences relative to the internal influences. Identifying and separating these factors is not trivial; however, efforts have been made by modeling electoral behavior with social influence factors [57]. The next step would be to study the changes detected here in more detail (for example, identify internal and external factors) or to explore local changes in periods of interest on a different time scale than the one considered in this study.

## Conclusion

A method was proposed to study the variability of votes in a parliament over time from the perspective of complex systems. The result and the proportion of agreement of each vote of the last 19 years of the Chilean Chamber of Deputies were considered. Two significant changes in temporal variability were found that may be directly related to major changes in the Chilean electoral system and the composition of the Chamber of Deputies. We argue that these methods help to detect changes that could be difficult to observe with traditional statistical methods.

## Supporting information

**S1 Appendix. Example of a Recurrence Plot (RP) for the 2006–2010 legislative period.** (DOCX)

**S2 Appendix. Composition of the chamber of deputies regarding what percentage is considered opposition and what percentage supports the government in each period.**
(DOCX)

**S3 Appendix. Detrended Fluctuation Analysis (DFA).**
(DOCX)

**S4 Appendix. Analyses to study if our data exhibits memory, regime shifts, and sensitive dependence on initial conditions.**
(DOCX)

## Author Contributions

**Conceptualization:** Diego Morales-Bader, Ramón D. Castillo, Ralf F. A. Cox.

**Data curation:** Diego Morales-Bader.

**Formal analysis:** Diego Morales-Bader, Ralf F. A. Cox.

**Funding acquisition:** Ramón D. Castillo.

**Investigation:** Diego Morales-Bader, Ramón D. Castillo, Ralf F. A. Cox, Carlos Ascencio-Garrido.

**Methodology:** Diego Morales-Bader, Ramón D. Castillo, Ralf F. A. Cox.

**Project administration:** Diego Morales-Bader.

**Software:** Diego Morales-Bader, Ralf F. A. Cox.

**Supervision:** Diego Morales-Bader, Ramón D. Castillo, Ralf F. A. Cox.

**Validation:** Diego Morales-Bader, Ramón D. Castillo, Ralf F. A. Cox, Carlos Ascencio-Garrido.

**Visualization:** Diego Morales-Bader, Ralf F. A. Cox.

**Writing – original draft:** Diego Morales-Bader, Ramón D. Castillo, Ralf F. A. Cox, Carlos Ascencio-Garrido.

**Writing – review & editing:** Diego Morales-Bader, Ramón D. Castillo, Ralf F. A. Cox, Carlos Ascencio-Garrido.

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
