## [Decision Letter · Decision Letter 0]

2 Oct 2022

PONE-D-22-21917Parliamentary roll-call voting as a complex dynamical system: The case of ChilePLOS ONE

Dear Dr. Castillo,

Thank you for submitting your manuscript to PLOS ONE. After careful consideration, we feel that it has merit but does not fully meet PLOS ONE’s publication criteria as it currently stands. Therefore, we invite you to submit a revised version of the manuscript that addresses the points raised during the review process.

We look forward to receiving your revised manuscript.

Best,

Prof. Dan Braha

Academic Editor

PLOS ONE

Journal Requirements:

Additional Editor Comments:

The authors note, correctly, that the results arise because of local interactions between the system’s components and global influences from its environment. The authors might be interested to examine and discuss the following work, which showed from both theoretical and empirical perspectives that this type of behavior is a general characteristic of complex dynamical systems in general and voting behavior in particular.

Braha, D., & De Aguiar, M. A. (2017). Voting contagion: Modeling and analysis of a century of US presidential elections. PloS one, 12(5), e0177970.

Reviewers' comments:

Reviewer's Responses to Questions

**Comments to the Author**

1. Is the manuscript technically sound, and do the data support the conclusions?

Reviewer #1: Partly

Reviewer #2: Yes

2. Has the statistical analysis been performed appropriately and rigorously? 

Reviewer #1: Yes

Reviewer #2: Yes

3. Have the authors made all data underlying the findings in their manuscript fully available?

Reviewer #1: No

Reviewer #2: Yes

4. Is the manuscript presented in an intelligible fashion and written in standard English?

Reviewer #1: Yes

Reviewer #2: No

5. Review Comments to the Author

Reviewer #1: This paper analyzes Chilean parliamentarian roll-call votes as a complex dynamic system. More specifically, two aspects are analyzed: the agreement ratio and the voting outcome (i.e., whether the vote was approved or not). In total, 19 years of data were considered, from 2002 to 2021. Several metrics were calculated to characterize the predictability and the dynamics of the system, including the recurrence rate, the determinism, the sample entropy and the DFA exponent. One objective of these analyzes is to verify if and to what extent the legislative work of the Chilean Chamber of Deputies presents characteristics of a complex system. A second objective is to identify relevant changes in the system dynamics and match these changes with socio-political and institutional transformations.

Regarding the first objective, this work succeeds in characterizing the dynamics of both the agreement ratio and the voting outcome according to a complex system. Results revealed some patterns, trends and changes in the behavior of both of these metrics over the 19 years analyzed. Comparing all results with a null model was an appropriate methodological decision. This comparison made it clear when the system approaches and differs from a random system. It is interesting to observe how the system can change drastically from one extreme to the other.

Regarding the second objective, some conjectures that help to explain some of the results were presented. For instance, the new electoral system in 2017 to explain why the agreement ratio became less predictable in the period 2018-2021 and the implementation of non-mandatory voting in 2013 to explain the increase in predictability of the vote outcome from 2014. While these analyzes shed some light on some of the big changes observed in the results, I would like more information about the more local changes. For instance, until 2014, all the three metrics shown in Fig 5 constantly cross the “random” line in the graph. Why is this happening? Are there particular categories of voting sessions that induce different behavior? Can these changes be explained by external events? Is there a particular political party responsible for dictating these changes?

In summary, my opinion on the second objective is that, while the high-level conjectures make sense, a deeper analysis of the motives involved in changes in system dynamics has been lacking. I feel that using the complex systems framework to analyze the agreement ratio and voting outcome of political systems is not fully justified in this article. In other words, why is this useful? How can these methods help us understand the dynamics of political systems? How do they correlate with the population's satisfaction with the current political system?

As another example, consider the the last paragraph of Page 23, where it is written:

“This similarity with the randomized series indicates a higher level of uncertainty. This is characteristic of systems transitioning to a new state or organization.”

We are in 2022. Has anything changed? I am not a specialist in Chilean politics, but I have read that recently “Chilean leftist Gabriel Boric won the country’s presidential runoff election on Sunday, capping a major revival for the country’s progressive left that has been on the rise since widespread protests roiled the Andean country two years ago.” Is there anything in the results that point to this change (and protests)?

Regarding the presentation of the paper, the text is well written and is easy to follow. Detailed references, including programming libraries, are provided for all methods. However, I missed the mathematical definition of metrics (e.g. forecast skill) for the article to be self-contained. I also suggest a revision in the paper to eliminate strong sentences that are not supported by references, such as “No one doubts that social and political systems have these characteristics.” Regarding the figures, one suggestion is to include vertical lines to mark the major changes in the Chilean political system, such as the implementation of non-mandatory voting in 2013.

Reviewer #2: Review for manuscript PONE-D-22-21917- Parliamentary roll-call voting as a complex dynamical system: The case of Chile.

By D. Morales-Bader et al.

Authors investigate the roll-call vote complex dynamic behavior in the particular case of the Chilean Chamber of Deputies. Research is carried out by analyzing the roll-call vote agreement ratio employing packages already implemented in popular programming languages. The roll-call vote agreement ratio is obtained from the free-available data provided by the aforementioned Lower House. Authors studied a period of time ranging from 2002 to 2021.

Nonlinear time series analysis of the agreement ratio shows that results of bills voted become more deterministic and recurrent. It is exposed that this characteristic begins to appear in 2013 when non-compulsory vote was introduced. However, the moving window technique shows that Lower House voting patterns became less predictable from 2018 to 2021 and it is argued that this feature is directly associated with a change introduced in 2017 in the way seats are allocated in Chilean parliament.

There are some issues in the manuscript that need to be covered. Commentaries by each section are listed as follows:

Generally speaking, the introduction section needs to be either re-organized or re-written in order to better motivate the main goal. The main idea of this research is to analyze roll-call vote data from the Chilean Chamber of Deputies using non-linear time series techniques. However, there are some paragraphs that deviate readers from this issue. Specifically, the fourth paragraph mentions that the scaling technique nominate cannot be used to analyze roll-call vote data because of its stochastic nature. Legislative systems cannot be considered as a stochastic system, instead, they belong to well-established deterministic ones. If you analyze the way how bills are introduced in the Lower Houses, beginning to be discussed in sub commissions until the plenary sessions, it is easy to perceive that the voting process defines the possible state of these bills at any time, and this is due because, the opinion that parliamentarians have to approve or disapprove a bill come from a deterministic process already introduced in the discussion and debates. The reason it is not possible to use scaling techniques in Legislative systems is because they are not scale-invariant. Paragraphs from sixth to eighth are out of scope. Authors persuade fractal properties and cite some examples in physiology. I did not find a plausible way to measure how much these passages contribute to roll-call vote analysis. In fact, establishing that Legislative systems can possess fractal properties is risky and can generate misunderstanding and concerns among readers. Previously, I mentioned that Legislative systems are not scale invariant, therefore, motivate roll-call vote analysis fractal-theory-based there make no sense. The possibility for using numerical methods based on the premises of complex systems and fractal theory does not imply that legislative data must be governed by these universal laws, it is just a mathematical artifact that allows us to identify some patterns that are associated with underlying political activities. No analogies can be established. By the way, Authors repeatedly mention a 1/f noise parameter but no description of what it is, what it measures and what can be in the political sense is carried out. Paragraphs from tenth to thirteenth paragraph are very focused on description of the numerical method to be used. This can be done in the methods section. Consider for example, replacing these paragraphs for one or two that motivate analysis of roll-call vote is done by using non-linear time series techniques and clarify that these methods are already implemented in popular programming languages in order to guarantee reproducibility.

For the methods section, the dataset sub--section needs to be complemented. This section does not mention the number of the Legislatures and name of chief executive branch for each period. Besides, there is no explicit mention of how many vote options a parliamentary has in order to express opinion for approving-disapproving bills. Are they nay, yea and abstention? What about absences, obstructions? Does the Chief of the Lower House have the ability to vote? This is important to better recognize possible political anomalies. Besides that, I was expecting some supplementary material exposing in detail, how data was collected, processed and transformed. The mathematical definition of agreement ratio is not easily readable, Website related to data was successfully verified and accessed. Paragraphs regarding description of numerical methods are well structured. It is sufficient to do some minor changes. My recommendation would be to write for each method a quick description of how it works, i.e, how the computation is carried out, what it is measured (already mentioned for all numerical methods used by Authors). Subsequently, mention the name of the package and programming language in which it is implemented. Finally, mention input parameters for computations, tolerance and convergence values if needed. There are some profound discussions presented in these sections that would be translated to the supplementary material.

The quality of figures in the results section are poor. Please, increase the size of axes labels, ticks and rotate all vertical labels for easy reading. Use coloured-style for figures. Introduce information of executive terms inside figures, for example by coloring delimited sections in figures 4 and 5. Captions for all figures are not self-explainable and are disconnected from the main text; it causes difficulties when the reader wants to analyze each figure in detail. I consider that figures 2, 4 and 5 must be the main figures for the manuscript. Remaining figures and all table-based results can be translated to the supplementary material. Respecting the text, structure of presentation makes reading uncomfortable because references for figures are made in an unordered way. Please correct this issue.

Regarding the discussion section, the second paragraph is confusing, Author argues that dynamics of the Chilean Lower House are similar to healthy physiological and behavioral systems. The question arising here is, was this the objective of this study? I consider that this paragraph is out of scope from the main idea. In the fourth paragraph, Authors again draws attention to the 1/f noise parameter. The only thing related to this parameter I found in the results section was some mention in the DFA technique. Using this method, they show two parameters for the agreement ratio and voting outcome time series, respectively, but no explicit functional expression or numerical result related to this parameter was found. Moreover, I consider it prudent to remove all discussion related to fractal-theory associations.

Finally, I consider the manuscript to be suitable for publication in PLOS ONE with a major revision taking into account, comments aforementioned exposed. Decision is based on the fact that this kind of work helps to explore new quantitative-based methods for improving political science. Besides that, Latin American legislative systems exhibit some particular emergence properties that are mandatory to study and characterize. In fact, the majority of these newbie features are not easy to visualize when studying, for example, the United States or United kingdom legislative branch. It is for that reason that this work is scientifically sound.

6. PLOS authors have the option to publish the peer review history of their article (what does this mean?). If published, this will include your full peer review and any attached files.

Reviewer #1: No

Reviewer #2: No

---

## [Author Response · Author response to Decision Letter 0]

13 Jan 2023

Response to Reviewers

Additional Editor Comments:

The authors note, correctly, that the results arise because of local interactions between the system’s components and global influences from its environment. The authors might be interested to examine and discuss the following work, which showed from both theoretical and empirical perspectives that this type of behavior is a general characteristic of complex dynamical systems in general and voting behavior in particular.

Braha, D., & De Aguiar, M. A. (2017). Voting contagion: Modeling and analysis of a century of US presidential elections. PloS one, 12(5), e0177970.

Response to editor

Many thanks for this suggestion, Braha and De Aguiar (2017) were incorporated into our manuscript 

Reviewer #1:

Comment 1

This paper analyzes Chilean parliamentarian roll-call votes as a complex dynamic system. More specifically, two aspects are analyzed: the agreement ratio and the voting outcome (i.e., whether the vote was approved or not). In total, 19 years of data were considered, from 2002 to 2021. Several metrics were calculated to characterize the predictability and the dynamics of the system, including the recurrence rate, the determinism, the sample entropy and the DFA exponent. One objective of these analyzes is to verify if and to what extent the legislative work of the Chilean Chamber of Deputies presents characteristics of a complex system. A second objective is to identify relevant changes in the system dynamics and match these changes with socio-political and institutional transformations. 

Response 1: This summary describes in a precise manner our work

Comment 2

Regarding the first objective, this work succeeds in characterizing the dynamics of both the agreement ratio and the voting outcome according to a complex system. Results revealed some patterns, trends and changes in the behavior of both of these metrics over the 19 years analyzed. Comparing all results with a null model was an appropriate methodological decision. This comparison made it clear when the system approaches and differs from a random system. It is interesting to observe how the system can change drastically from one extreme to the other. 

Response 2:

We agree with this comment in which we compared the original versus the shuffled times series. The original time series shows shifts and drastic fluctuations similar to chaotic systems. In appendix A, we incorporate a new figure in which recurrent plots of the original agreement ratio and its respective random shuffle are compared (pp 29).

Comment 3

Regarding the second objective, some conjectures that help to explain some of the results were presented. For instance, the new electoral system in 2017 to explain why the agreement ratio became less predictable in the period 2018-2021 and the implementation of non-mandatory voting in 2013 to explain the increase in predictability of the vote outcome from 2014. While these analyzes shed some light on some of the big changes observed in the results, I would like more information about the more local changes. For instance, until 2014, all the three metrics shown in Fig 5 constantly cross the “random” line in the graph. Why is this happening? Are there particular categories of voting sessions that induce different behavior? Can these changes be explained by external events? Is there a particular political party responsible for dictating these changes?

Response 3

Given the time scale we are using in this study, it is challenging to establish associations of small events (compared to our time scale) with significant impacts. Several evident local changes may be due to natural voting behavior (e.g., the rise and fall of metrics), while a multiplicity of factors may cause some. For this reason, it is challenging to associate events with minor changes. For this, in the manuscript, we recommend using these methods on a different time scale, for example, focusing only on a legislative period.

In the manuscript, we include a discussion of this apparent local change before the significant change detected in 2014. First, in the period prior to the change (2010-2014), we found that that was the period with the highest percentage of opposition deputies, while the following period (2014-2018) was the period with the lowest percentage of opposition. However, this may not explain the variation detected before the 2014 change. One hypothesis we discuss in the new version of the article is that this variation close to the random line may represent an early warning signal prior to a phase change or transition, which is also often detected in physical and biological systems.

While these hypotheses are discussed in the manuscript, the final recommendation in order to answer the questions raised is to study in more detail this period with a focus on local changes."

Comment 4

In summary, my opinion on the second objective is that, while the high-level conjectures make sense, a deeper analysis of the motives involved in changes in system dynamics has been lacking. I feel that using the complex systems framework to analyze the agreement ratio and voting outcome of political systems is not fully justified in this article. In other words, why is this useful? How can these methods help us understand the dynamics of political systems? How do they correlate with the population's satisfaction with the current political system? As another example, consider the the last paragraph of Page 23, where it is written: “This similarity with the randomized series indicates a higher level of uncertainty. This is characteristic of systems transitioning to a new state or organization.” We are in 2022. Has anything changed? I am not a specialist in Chilean politics, but I have read that recently “Chilean leftist Gabriel Boric won the country’s presidential runoff election on Sunday, capping a major revival for the country’s progressive left that has been on the rise since widespread protests roiled the Andean country two years ago.” Is there anything in the results that point to this change (and protests)? Regarding the presentation of the paper, the text is well written and is easy to follow. Detailed references, including programming libraries, are provided for all methods. However, I missed the mathematical definition of metrics (e.g. forecast skill) for the article to be self-contained. I also suggest a revision in the paper to eliminate strong sentences that are not supported by references, such as “No one doubts that social and political systems have these characteristics.” Regarding the figures, one suggestion is to include vertical lines to mark the major changes in the Chilean political system, such as the implementation of non-mandatory voting in 2013. 

Response 4

We agree that in the document's first version, the justification and rationale for applying the methods were unclear. This new version reinforces this point with a logical order and more precise wording.

These methods aim to detect significant changes in the temporal variability of voting. While votes always vary over time due to internal and external factors, the usefulness of these methods is that they help us detect when a time series transitions to a state different from this usual variability. This change in variability may not have to do with a change in the average or other measures of central tendency but may only be discovered when we incorporate time as a variable. While with other methods, this variability can be considered as "noise" to be eliminated, we study this "noise" with these methods.

While the analyses applied in this study do not allow us to establish causal associations, they help us focus on where to guide future studies. Now that we have detected significant global changes, we can focus on looking for local changes or try to associate the changes detected here with other variables.

Strong sentences were eliminated, and mathematical definitions of the methods used were added.

Reviewer #2:

Review for manuscript PONE-D-22-21917- Parliamentary roll-call voting as a complex dynamical system: The case of Chile.

Comment 1

Authors investigate the roll-call vote complex dynamic behavior in the particular case of the Chilean Chamber of Deputies. Research is carried out by analyzing the roll-call vote agreement ratio employing packages already implemented in popular programming languages. The roll-call vote agreement ratio is obtained from the free-available data provided by the aforementioned Lower House. Authors studied a period of time ranging from 2002 to 2021. 

Response 1

We agree with this description

Comment 2

Nonlinear time series analysis of the agreement ratio shows that results of bills voted become more deterministic and recurrent. It is exposed that this characteristic begins to appear in 2013 when non-compulsory vote was introduced. However, the moving window technique shows that Lower House voting patterns became less predictable from 2018 to 2021 and it is argued that this feature is directly associated with a change introduced in 2017 in the way seats are allocated in Chilean parliament. 

Response 2

We agree with this summary; in this new version, we provide more details about the analyses.

Comment 3

There are some issues in the manuscript that need to be covered. Commentaries by each section are listed as follows: 

Generally speaking, the introduction section needs to be either re-organized or re-written in order to better motivate the main goal. The main idea of this research is to analyze roll-call vote data from the Chilean Chamber of Deputies using non-linear time series techniques. However, there are some paragraphs that deviate readers from this issue. Specifically, the fourth paragraph mentions that the scaling technique nominate cannot be used to analyze roll-call vote data because of its stochastic nature. Legislative systems cannot be considered as a stochastic system, instead, they belong to well-established deterministic ones. If you analyze the way how bills are introduced in the Lower Houses, beginning to be discussed in sub commissions until the plenary sessions, it is easy to perceive that the voting process defines the possible state of these bills at any time, and this is due because, the opinion that parliamentarians have to approve or disapprove a bill come from a deterministic process already introduced in the discussion and debates. The reason it is not possible to use scaling techniques in Legislative systems is because they are not scale-invariant. Paragraphs from sixth to eighth are out of scope. Authors persuade fractal properties and cite some examples in physiology. I did not find a plausible way to measure how much these passages contribute to roll-call vote analysis. In fact, establishing that Legislative systems can possess fractal properties is risky and can generate misunderstanding and concerns among readers. Previously, I mentioned that Legislative systems are not scale invariant, therefore, motivate roll-call vote analysis fractal-theory-based there make no sense. The possibility for using numerical methods based on the premises of complex systems and fractal theory does not imply that legislative data must be governed by these universal laws, it is just a mathematical artifact that allows us to identify some patterns that are associated with underlying political activities. No analogies can be established. By the way, Authors repeatedly mention a 1/f noise parameter but no description of what it is, what it measures and what can be in the political sense is carried out. Paragraphs from tenth to thirteenth paragraph are very focused on description of the numerical method to be used. This can be done in the methods section. Consider for example, replacing these paragraphs for one or two that motivate analysis of roll-call vote is done by using non-linear time series techniques and clarify that these methods are already implemented in popular programming languages in order to guarantee reproducibility. 

Response 3

We agree with these suggestions, and the text was reorganized according to the reviewer's comments. A paragraph was added indicating that although voting is deterministic because we can even know in advance how several parliamentarians are going to vote on a specific issue, we argue that by including the time variable, voting is no longer as deterministic because many more variables come into play.

References to fractal properties, 1\\f noise, and Detrended Fluctuation Analysis (DFA) were removed. The DFA was moved to the annexes since it may interest other researchers who want to compare their results with those we found. DFA can be useful for comparing different signals or processes, which was not the study's primary goal. In the context of our study, we used it to explore the self-similarity and temporal stability of the data. We now only mention DFA once in the discussion, citing where it is located in the appendix.

Most of the discussion of numerical methods was moved to the method section, leaving only the rationale for choosing these methods in the introduction.

Comment 4

For the methods section, the dataset sub--section needs to be complemented. This section does not mention the number of the Legislatures and name of chief executive branch for each period. Besides, there is no explicit mention of how many vote options a parliamentary has in order to express opinion for approving-disapproving bills. Are they nay, yea and abstention? What about absences, obstructions? Does the Chief of the Lower House have the ability to vote? This is important to better recognize possible political anomalies. Besides that, I was expecting some supplementary material exposing in detail, how data was collected, processed and transformed. The mathematical definition of agreement ratio is not easily readable, Website related to data was successfully verified and accessed. Paragraphs regarding description of numerical methods are well structured. It is sufficient to do some minor changes. My recommendation would be to write for each method a quick description of how it works, i.e, how the computation is carried out, what it is measured (already mentioned for all numerical methods used by Authors). Subsequently, mention the name of the package and programming language in which it is implemented. Finally, mention input parameters for computations, tolerance and convergence values if needed. There are some profound discussions presented in these sections that would be translated to the supplementary material.

Response 4

More information about the Chilean political system and the functioning of the Chamber of Deputies was added in the introduction, covering much of what was consulted by the reviewer.

More information on data collection and processing was added. A Jupyter Notebook was made available to download and organize the data from the Chilean congressional website. This notebook is also used to process the data, for example, to eliminate or fill in missing data.

Comment 5

The quality of figures in the results section are poor. Please, increase the size of axes labels, ticks and rotate all vertical labels for easy reading. Use coloured-style for figures. Introduce information of executive terms inside figures, for example by coloring delimited sections in figures 4 and 5. 

Captions for all figures are not self-explainable and are disconnected from the main text; it causes difficulties when the reader wants to analyze each figure in detail. I consider that figures 2, 4 and 5 must be the main figures for the manuscript. Remaining figures and all table-based results can be translated to the supplementary material. Respecting the text, structure of presentation makes reading uncomfortable because references for figures are made in an unordered way. Please correct this issue. 

Response 5

The figures were re-created to make their interpretation understandable. The issues were corrected according to the reviewer's suggestions

Comment 6

Regarding the discussion section, the second paragraph is confusing, Author argues that dynamics of the Chilean Lower House are similar to healthy physiological and behavioral systems. The question arising here is, was this the objective of this study? I consider that this paragraph is out of scope from the main idea. In the fourth paragraph, Authors again draws attention to the 1/f noise parameter. The only thing related to this parameter I found in the results section was some mention in the DFA technique. Using this method, they show two parameters for the agreement ratio and voting outcome time series, respectively, but no explicit functional expression or numerical result related to this parameter was found. Moreover, I consider it prudent to remove all discussions related to fractal-theory associations. 

Response 6

We eliminate the second paragraph in the discussion and any reference to 1/f-pink noise. However, in the appendix section, readers can see DFA results and an explanation of what DFA means when it is applied to roll-call voting processes.

Comment 7

Finally, I consider the manuscript to be suitable for publication in PLOS ONE with a major revision taking into account, comments aforementioned exposed. Decision is based on the fact that this kind of work helps to explore new quantitative-based methods for improving political science. Besides that, Latin American legislative systems exhibit some particular emergence properties that are mandatory to study and characterize. In fact, the majority of these newbie features are not easy to visualize when studying, for example, the United States or United kingdom legislative branch. It is for that reason that this work is scientifically sound.

Response 7

According to the reviewers’ suggestions and comments we made major revisions to our original manuscr

---

## [Editor Report · Decision Letter 1]

19 Jan 2023

PONE-D-22-21917R1Parliamentary roll-call voting as a complex dynamical system: The case of ChilePLOS ONE

Dear Dr. Castillo,

Thank you for submitting your manuscript to PLOS ONE. After careful consideration, we feel that it has merit but does not fully meet PLOS ONE’s publication criteria as it currently stands. Therefore, we invite you to submit a revised version of the manuscript that addresses the points raised during the review process. Academic Editor Comment:

The authors effectively responded to the referees' comments. The authors state, "... Braha and De Aguiar (2017) were incorporated into our manuscript"; however, this citation is absent from both the reference list and the main text. In addition, the presented methods can account for the observed data variability. Here, some discussion would be useful.

We look forward to receiving your revised manuscript.

Kind regards,

Dan Braha

Academic Editor

PLOS ONE

Journal Requirements:

Additional Editor Comments:

The authors effectively responded to the referees' comments. The authors state, "... Braha and De Aguiar (2017) were incorporated into our manuscript"; however, this citation is absent from both the reference list and the main text. In addition, the presented methods can account for the observed data variability. Here, some discussion would be useful.

---

## [Author Response · Author response to Decision Letter 1]

31 Jan 2023

Response to Reviewers.

Comment: The authors effectively responded to the referees' comments. The authors state, "... Braha and de Aguiar (2017) were incorporated into our manuscript"; however, this citation is absent from both the reference list and the main text. In addition, the presented methods can account for the observed data variability. Here, some discussion would be useful.

Response: The reference to Braha and de Aguiar (2017) corresponds to reference number 55; however, it was erroneously cited. In the version we are sending now, we have corrected this error. We have also rephrased the paragraph where this reference was cited (lines 523 to 532).

Paragraph in previous version:

A limitation of this study is that, given its descriptive approach, it is impossible to establish direct relationships between national policy events and the changes in our data. We can only make conjectures given the association of these changes' timing with the several events' timing and variables can co-occur with internal and external influencing factors [55]. Identifying and separating these factors is not trivial. The next step would be to study the

changes detected here in more detail or to explore local changes in periods of interest on a

different time scale than the one considered in this study.

Paragraph in this version:

A limitation of this study is that, given its descriptive approach, it is impossible to establish direct relationships between national policy events and the changes found in this study. We can only make conjectures, given the association of these changes' timing with the events' timing. Several events and variables may co-occur with internal and external influencing factors. These factors influence in different ways depending on the strength of the external influences relative to the internal influences. Identifying and separating these factors is not trivial; however, efforts have been made by modeling electoral behavior with social influence factors [55]. The next step would be to study the changes detected here in more detail (for example, identify internal and external factors) or to explore local changes in periods of interest on a different time scale than the one considered in this study.

Reference number 55 in previous version:

[55] Aguilera M, Morer I, Barandiaran X, Bedia M. Quantifying Political Self-Organization in Social Media. Fractal patterns in the Spanish 15M movement on Twitter. Adv. Artif. Life ECAL 2013, MIT Press; 2013, p. 395–402. https://doi.org/10.7551/978-0-262-31709-2-ch057.

Reference number 55 in this version:

[55] Braha D, de Aguiar MAM. Voting contagion: Modeling and analysis of a century of U.S. presidential elections. PLOS ONE 2017;12:e0177970. https://doi.org/10.1371/journal.pone.0177970.

---

## [Editor Report · Decision Letter 2]

2 Feb 2023

Parliamentary roll-call voting as a complex dynamical system: The case of Chile

PONE-D-22-21917R2

Dear Dr. Castillo,

We’re pleased to inform you that your manuscript has been judged scientifically suitable for publication and will be formally accepted for publication once it meets all outstanding technical requirements.

Kind regards,

Dan Braha

Academic Editor

PLOS ONE

Additional Editor Comments (optional):

The authors made the necessary modifications. I would recommend accepting it as-is for publication.
---

## [Editor Report · Acceptance letter]

21 Feb 2023

PONE-D-22-21917R2 

Parliamentary roll-call voting as a complex dynamical system: The case of Chile 

Dear Dr. Castillo:

I'm pleased to inform you that your manuscript has been deemed suitable for publication in PLOS ONE. Congratulations! Your manuscript is now with our production department. 

Kind regards, 

on behalf of

Professor Dan Braha 

Academic Editor

PLOS ONE